SHORT REPORT                                                              Open Access

# Minimal genome-wide human CRISPR-Cas9 library

Emanuel Gonçalves[1], Mark Thomas[1], Fiona M. Behan[1], Gabriele Picco[1], Clare Pacini[1,2], Felicity Allen[1], Alessandro Vinceti[3], Mamta Sharma[1], David A. Jackson[1], Stacey Price[1], Charlotte M. Beaver[1], Oliver Dovey[1], David Parry-Smith[1], Francesco Iorio[1,3], Leopold Parts[1,4], Kosuke Yusa[5] and Mathew J. Garnett[1*]

* Correspondence: mg12@sanger.ac.
uk
[1]Wellcome Sanger Institute,
Wellcome Genome Campus,
Hinxton, UK
Full list of author information is
available at the end of the article

## Abstract

CRISPR guide RNA libraries have been iteratively improved to provide increasingly
efficient reagents, although their large size is a barrier for many applications. We
design an optimised minimal genome-wide human CRISPR-Cas9 library (MinLibCas9)
by mining existing large-scale gene loss-of-function datasets, resulting in a greater
than 42% reduction in size compared to other CRISPR-Cas9 libraries while preserving
assay sensitivity and specificity. MinLibCas9 provides backward compatibility with
existing datasets, increases the dynamic range of CRISPR-Cas9 screens and extends
their application to complex models and assays.

**Keywords:** CRISPR-Cas9, Genome-wide, Minimal library, Organoid, KS score

## Main text

CRISPR-Cas9 loss-of-function screens have been used in a variety of model organisms,
including human cells [1, 2]. All broadly used Cas9 genome-wide libraries have at least
4 single-guide RNA (sgRNA) per gene and contain over 65,000 sgRNAs [2–13] (Fig. 1a)
(Additional file 2: Table S1). In silico down-sampling analyses have shown that 2
sgRNAs per gene can recover previously defined essential genes, a key quality control
measure [12, 14] (Additional file 1: Fig. S1). Genome-wide CRISPR-Cas9 sgRNA librar-
ies have been iteratively optimised to reduce off-target activity and increase on-target
efficiency [8, 9, 11, 12, 15, 16], mostly using nucleotide sequence-based sgRNA efficacy
prediction algorithms [9, 11, 15, 16]. The recent availability of data from CRISPR-Cas9
knockout screens performed in hundreds of cell lines [13, 17] makes it now possible to
empirically improve library design through selection of sgRNA with strong and con-
sistent biological effects across diverse contexts, incorporating additional factors that
might influence guide efficacy [18, 19]. Smaller genome-wide CRISPR libraries are
more cost-effective and increase feasibility when assaying complex models (e.g. primary
cultures, organoids, co-cultures, in vivo screens), measuring complex phenotypic end-
points (e.g. scRNAseq or perturbations), and when probing genetic interactions using
multiplexed CRISPR-Cas9 libraries. We therefore assembled a standardised resource

**Fig. 1** Genome-wide human CRISPR-Cas9 sgRNA libraries. **a** Number of sgRNAs in each CRISPR-Cas9 library since the first reported genome-wide screens, excluding Wang et al. [3] which targets 7114 genes. **b** Area under the recall curve of sgRNAs targeting known essential ($n = 1469$) and non-essential ($n = 3251$) genes, and non-targeting guides ($n = 997$). Recall curves were calculated for each replicate of Project Score [13] ($n = 663$) and represented by the cumulative distribution of each sgRNA group across all sgRNAs sorted by ascending fold-changes. Box-and-whisker plots show 1.5× interquartile ranges and 5–95th percentiles, centres indicate medians. **c** Fold-change distribution, based on Project Score data-set of the different sgRNAs groups. Diagram depicting how the KS scores vary across the CRISPR-screens fold-change range. KS scores are calculated by testing if the distribution of each sgRNA across cell lines is drawn from that of the non-targeting sgRNAs using a two-sided Kolmogorov-Smirnov distribution

that harnesses large-scale CRISPR-Cas9 screens, together with multiple efficiency metrics to evaluate and rank over 300,000 unique sgRNAs originating from the most broadly adopted libraries. These were used to design an optimised minimal genome-wide library with two sgRNAs per gene (MinLibCas9) that allows for backward compatibility with large resources of CRISPR-Cas9 screens of cancer cell models.

We began by compiling multiple genome-wide CRISPR-Cas9 sgRNA libraries, namely Project Score (Kosuke Yusa V1.1), Avana, Brunello and TKOv3 [8, 9, 11–13], to provide standardised annotation for 300,167 unique sgRNAs with a median of 19 sgRNA per gene (Additional file 3: Table S2). This included updated mapping to GRCh38 and off-target summaries using the CRISPR genome editing database WGE [20], and where possible multiple guide efficacy metrics (JACKS [14], Rule Set 2 [9], FORECasT [21] and scores exported from CRISPOR [22] such as MIT specificity [23] and CrisprScan [24]). This reference library provides a single standardised resource to select guides based on multiple user-defined criteria.

We assembled a minimal genome-wide human CRISPR-Cas9 library (MinLibCas9) through multiple iterative design steps (Additional file 1: Fig. S2a and Additional file 4: Table S3). Preference was given to guides from the Project Score or Avana libraries, as these have been robustly validated and empirically tested across hundreds of cancer cell lines. Additionally, we prioritised Project Score guides [13] to preserve library consistency [25] and to allow in silico down-sampling benchmarking of MinLibCas9. We first minimised potential sgRNA off-target activities. Updated sgRNA off-target summaries in the reference library were used to exclude non-selective guides [20]. In addition, JACKS scores [14] were used to identify sgRNAs with fitness profiles dissimilar to the mean of all sgRNAs targeting the same gene, thereby empirically excluding sgRNAs with outlier profiles suggestive of off-target or reduced on-target activity.

We then sought to prioritise guides with maximal on-target activity. Approximately one third of all human protein-coding genes can induce a cellular loss-of-fitness effect upon knockout in cancer cell lines [13, 17]; thus, for the remaining two thirds, it is challenging to distinguish between efficient and non-efficient targeting sgRNAs. The introduction of CRISPR-Cas9-mediated DNA double-strand breaks induces a weak loss-of-fitness effect in cells regardless of the targeted site or gene [26–28]. The Project Score library included 997 non-targeting sgRNAs that do not align to any region in the human genome [13]. These non-targeting sgRNAs were positively enriched across all samples, which demonstrates a detectable relative growth advantage in the absence of a DNA double-stranded break (Fig. 1b, c). Thus, to empirically identify optimal on-target sgRNAs, we performed a non-parametric Kolmogorov-Smirnov test (KS score) comparing the distribution of the fitness fold-changes of every sgRNA to that of the non-targeting guides (Fig. 1c). Guides with high KS scores (values closer to 1) have strong negative or positive median fold-changes, whereas those with low KS scores are more likely to have weak or no activity, similar to non-targeting controls (Additional file 1: Fig. S2b). Thus, the KS score assigns an emperical estimate of sgRNA efficiency, even for guides that target genes which were not required for cell fitness. We expanded this approach and estimated KS scores for all sgRNAs in the Avana library (DepMap19Q2 release) [17, 29, 30].

No strong association between different sgRNA design metrics (i.e. KS, JACKS [14], Rule Set 2 [9] and FORECasT percentage of in-frame deletions [21]) was observed, suggesting that they inform on different aspects of guide efficiency and provide complementary information (Additional file 1: Fig. S2b). A strategy combining JACKS to exclude guides with outlier effects followed by ranking the remaining guides using KS scores improved recall rates of gene dependencies identified with the original library (Additional file 1: Fig. S3). Notably, the top 2 selected sgRNAs using this approach performed similarly compared to the complete library, and limited improvement was observed when considering more than 2 guides per gene. Based on these selection criteria, we designed MinLibCas9 which targets 18,761 genes using 2 optimal sgRNAs per gene and has a total of 37,522 gene-targeting and 200 non-targeting sgRNAs (Additional file 4: Table S3).

To independently validate our guide selection procedure, we used sgRNA selectivity (MIT specificity [23]) and efficiency (CrisprScan [24]) metrics from CRISPOR [22] that were not used for library design. MinLibCas9 outperformed or had similar scores to existing libraries (Additional file 1: Fig. S4a). Poly-T stretches ≥ 4 were largely absent from the reference library and T-stretches ≤ 4 displayed minimal to no impact on guide expression and efficacy (Additional file 1: Fig. S4b). Overall, MinLibCas9 library targets an additional 964 protein-coding genes compared to the original Project Score library. Notably, it is 41.7% to 79.9% smaller in size compared to any currently publicly available genome-wide CRISPR-Cas9 human library, and specifically 62.7% and 66.7% smaller than Project Score and Avana libraries, respectively (Fig. 1a).

Prioritising the selection of the Project Score library guides ensures that we could in silico benchmark our MinLibCas9 by subsampling the full set of sgRNAs across 245 cancer cell lines (90.6% of the sgRNAs originated from Project Score library). Overall, MinLibCas9 preserved the ability to identify known essential genes (Fig. 2a) and recovered the majority of significant dependencies found with the Project Score library, with an average precision greater than 89.8% in at least 80% of the 245 cancer cell lines

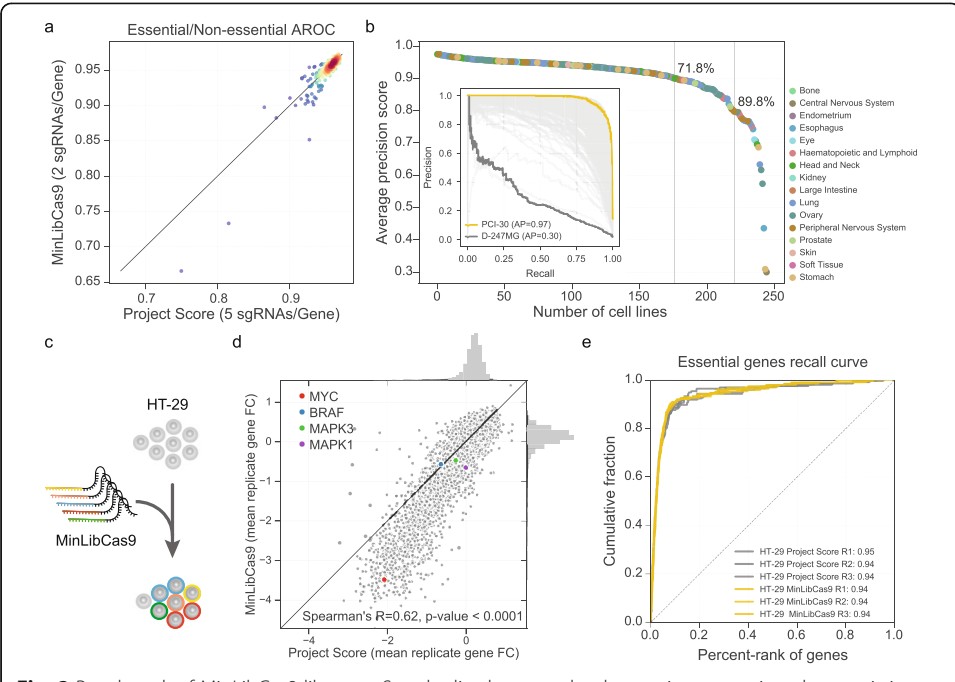

**Fig. 2** Benchmark of MinLibCas9 library. **a** Standardised area under the receiver operating characteristic curve for 245 cell lines at 20% FDR for the essential genes calculated using the minimal and original full library. **b** Average Precision (AP) scores to classify significant gene dependencies identified at 1% FDR in Project Score library using gene fold-changes from MinLibCas9. Recall-Precision curves for all cell lines are represented in the inset and cell lines with the lowest and highest AP score are highlighted. **c**, CRISPR-Cas9 screen performed on HT-29 cancer cells using MinLibCas9 library. **d** Correlation between gene-level fold-changes obtained with the two libraries. **e** Recall of essential genes across all replicates for HT-29 performed with MinLibCas9 and Project Score libraries

(Fig. 2b, Additional file 1: Fig. S4c). In the few instances where lower precision was observed, this was associated with lower data quality (Additional file 1: Fig. S4d). The selected optimal two guides provided gene-level fold-changes largely concordant with the original library (mean Spearman's $R = 0.77$) and a greater fold-change dynamic range (Additional file 1: Fig. S4e). The cumulative number of significantly dependent cell lines identified per gene was well correlated (Spearman's $R = 0.88$, $p$ value < 0.001) (Additional file 1: Fig. S4f). Dependencies not identified with MinLibCas9 had on average weaker fold-changes (two-sided Welch's $t$ test $p$ value < 0.001) (Additional file 1: Fig. S4g). A total of 107 genes had discordant significant dependencies, primarily due to sub-groups of sgRNAs with very distinct fold-change profiles. These guides were therefore replaced with others from the reference library as a final design step (Additional file 1: Fig. S4f). Different levels of sgRNA coverage ($\times 25$, $\times 50$, $\times 75$, $\times 100$ and $\times 500$) during screening revealed, for both the minimal and full library, lower coverage has no impact on identifying essential genes (Additional file 1: Fig. S5a). Replicate correlation at the gene-level was lower with the minimal library and is to be expected when considering the lower number of sgRNAs per gene (Additional file 1: Fig. S5b).

To further benchmark MinLibCas9, we assessed if it could recapitulate dependencies in more complex models and assays independent from those used for library design. We began by analysing CRISPR-Cas9 screens used to identify genes that enhance or suppress sensitivity to a BRAF inhibitor (dabrafenib) in a partially-sensitive BRAF-mutant colorectal cancer cell line (HT-29) (Additional file 1: Fig. S6a). Using in silico

down-sampling analysis, gene fold-changes with both libraries were strongly correlated (Spearman's $R = 0.72$, $p$ value $< 0.001$) and the time-series profiles of top dependencies were consistently identified (Additional file 1: Fig. S6b and S6c). We also performed genome-wide screens in three 3D organoid cultures and confirmed that gene fold-changes between the Project Score and in silico down-sampled MinLibCas9 were strongly correlated (average Spearman's $R = 0.70$), confirming the minimal library provides similar replicates correlation and capacity to identify known essential genes (Additional file 1: Fig. S7).

Finally, we synthesised and cloned the final MinLibCas9 library and re-screened the HT-29 colorectal cancer cell line (Fig. 2c, Additional file 1: Fig. S8a and S8b). Compared to screens performed with Project Score library [13], we observed good correlation between gene-level fold-changes (Fig. 2d, Additional file 1: Fig. S8c) and similar recall of essential genes (Fig. 2e, Additional file 1: Fig. S8d). Consistent with the subsampling analysis, MinLibCas9 showed an increased dynamic range with stronger overall fold-changes, which improved the identification of previous cancer dependencies for this particular cell line, including MYC copy-number amplification, BRAF V600E gain-of-function mutation and ERK1/2 (MAPK1/2) sensitivity to drug inhibition [31–33] (Fig. 2d and Additional file 1: Fig. S8e). Lastly, we confirmed that CRISPR-Cas9 analytical tools used to correct copy number deleterious biases, CRISPRcleanR [34] and Crispy [28], and call significant gene dependencies in cells, BAGEL [35], can be readily utilised to analyse MinLibCas9 screens (Additional file 1: Fig. S9). These results confirm that MinLibCas9 can identify cancer cell dependencies, provides improved dynamic range for detecting cancer vulnerabilities and is compatible with existing CRISPR-Cas9 analytical tools and pipelines.

In summary, we designed an optimised minimal genome-wide human CRISPR-Cas9 library (MinLibCas9) using previously reported experimental data to select and rank sgRNAs and validated its utility in multiple experimental settings. MinLibCas9 is at least 42% smaller than most currently used libraries and preserves the sensitivity and specificity required to identify gene dependencies. Prioritising Project Score library sgRNAs ensures backward compatibility to a high quality and extensively validated library and mitigates library-specific batch effects allowing a direct integration with hundreds of already performed screens. Furthermore, our reference guide library, with comprehensive and standardised efficiency metrics for 300,167 unique sgRNA from the most broadly accepted libraries, is a valuable resource to support user-defined selection of optimised CRISPR-Cas9 reagents. Underlining the importance of smaller CRISPR libraries, recent studies have proposed to either exploit large-scale CRISPR-Cas9 datasets to empirically improve library design [36], build smaller complementary libraries from a single genome-wide library [12, 37] or by utilising multiplexed Cas9 [38] and Cas12a [39, 40] systems. Uniquely, MinLibCas9 combines some of these strategies to design a minimal and optimised sgRNA library, which unlocks the application of Cas9 genome-wide screens to complex models currently limited to the delivery of libraries focused on predefined and small gene sets. Moreover, it provides a data-driven approach to prioritise the selection of the most effective sgRNAs for assays using more complex read-outs, e.g. Perturb-seq [41, 42], and to build large-scale genetic interaction libraries.

## Methods

### CRISPR-Cas9 screens analysis

Screen analysis started with the sgRNA read count matrices. Guides with less than 30 counts in the control condition, i.e. plasmid DNA (pDNA), were excluded. Read counts were normalised to reads per million $(G')$ within each sample using the following formula:

$$G'_i = \left(G_i / \sum_j^n G_j\right) \times 10^6$$

where $G_i$ represents the raw counts of sgRNA i. A pseudo count of 1 was added to the whole matrix and $\log_2$ fold-changes were then calculated compared to pDNA. sgRNAs recall curves are drawn by sorting the guides by fold-change, from the most negative to the most positive, and then the cumulative distribution is calculated for the different guide groups (i.e. targeting essential genes, targeting non-essential genes and non-targeting sgRNAs). Next, the area under the recall curve is calculated, which represents the enrichment of each group towards negative or positive fold-changes; being an area of 0.5 the random expectation. sgRNA down-sampling analyses were performed by randomly sampling $n$ sgRNAs without replacement.

Gene-level fold-changes were calculated by grouping all sgRNAs by their targeting gene and taking the mean of the fold-changes. Similarly, replicates of the same cell line were mean-averaged. Gene dependencies were defined as significant, on a per-sample basis, if the gene $\log_2$ fold-change was lower than the fold-change threshold at which essential genes were found at 1% false discovery rate (FDR) from non-essential genes in the receiver operating characteristic (ROC) curve [6, 30]. Similarly to Allen et al. [14], the same ROC curve was used to estimate the performance of the sample to recapitulate previously defined essential genes by taking the area under the ROC (AROC) curve at 20% FDR, i.e. standardised partial AUC at maximum 20% false positive rate.

Recall-Precision curves of gene dependencies were drawn for each cell line by taking the significant gene dependencies (1% FDR) identified with the Project Score library and using the gene fold-changes obtained with MinLibCas9. Curves were summarised using average precision (AP) scores, defined as follows:

$$AP = \sum_j^n (R_j - R_{j-1}) P_j$$

where $P_n$ and $R_n$ are the precision and recall at the $n$th threshold. AP score is a similar metric to the area under the Precision-Recall curve.

### Copy number correction and significant dependencies analyses

Raw counts for the HT-29 Project Score library screens were downloaded from the Project Score website (https://score.depmap.sanger.ac.uk/). sgRNAs with less than 30 reads in the plasmid were removed. Read counts were corrected for copy number deleterious bias on a per replicate basis using two approaches: (i) unsupervised using CRISPRcleanR [34] with default parameters, and (ii) supervised, by providing copy number segments, using Crispy with the minimum number of sgRNA per segment set to 4. The CRISPRcleanR corrected fold changes were then processed using BAGELR [13, 35] with 2000 bootstrap iterations.

## Guide efficacy KS-score

CRISPR-Cas9 sgRNAs Kolmogorov-Smirnov scores (KS scores) is a two-sided test assessing if the sgRNA fold-changes and the median fold-changes of all non-targeting sgRNAs are drawn from the same distribution (function ks_2samp from scipy [43] Python package was used). KS scores range between 0 and 1, and values closer to 0 represent sgRNAs with a distribution similar to non-targeting sgRNAs, whereas values closer to 1 represent the most dissimilar sgRNAs. KS scores were estimated for 100,262 sgRNAs across 663 samples (245 unique cancer cell lines) of Project Score data-set [13] and for 73,911 sgRNAs across 1257 samples (562 unique cancer cell lines) of the Broad DepMap19Q2 data-set [17, 29] (Additional file 5: Table S4).

## Reference master CRISPR-Cas9 library

All the sgRNAs described in the Project Score [13], Avana [9], Brunello [9, 12] and TKOv3 [11] libraries were assembled into a single reference master library containing a total of 354,715 sgRNAs with a median of 19 guides per gene (Additional file 3: Table S2). The location of each guide and PAM on the GRCh37 assembly was confirmed for the Project Score, Avana and TKoV3 libraries using CasOffinder [44], before being lifted-over to the GRCh38 assembly. Unique CRIS PR_IDs and off-target summaries were extracted from the WGE CRISPR database [20] for all sgRNAs using the GRCh38 genomic coordinates and confirmed by sequence identity, with any inconsistent matches manually verified. Where possible, guides were complemented with efficiency scores from Rule Set 2 [9, 15], JACKS [14] and FORECasT in-frame indels [21]. Guides from Avana and Project Score libraries were also annotated with KS scores estimated from large-scale screens available for each library [13, 17, 29]. Additionally, CRISPOR resource [22] was used to import off-target MIT-specificity scores [23] and on-target efficiency CRIS PR-scan scores [24]. All CRISPOR scores were extracted from the crisprAllTargets dataset in the UCSC genome browser using the TableBrowser function. Overlap between the guides and Pfam protein domains [45] was also determined using transcript mappings extracted from the ucscGenePfam dataset.

Although ≥ 4 poly-T stretched can act as RNA Polymerase (Pol III) terminators for the often used U6 promoter [46], we determined that the majority of sgRNAs containing poly-T stretches of 4 and 5 (no sgRNA in the master library had more than 5 long T-stretches) are in fact expressed. Most poly-T containing sgRNAs display consistent effects with other non-poly-T containing sgRNAs targeting the same gene and have good sgRNA metrics. The effect varies depending on the position of the poly-T within the sgRNA, with decreased efficiencies observed closer to the PAM motif (Additional file 1: Fig. S4b). This is consistent with previous studies, which specify that full termination is only achieved with a poly-T stretch ≥ 6 [46]. Furthermore, substitution of the first base of the sgRNA to a G is recommended to improve guide expression [16] and therefore reduces the number of 4 T-stretches and no 5 T-stretch is considered in the reference library. Annotation is provided if a 4 or 5 poly-T stretch is contained within the sgRNA sequence (excluding PAM motif).

### CRISPR-Cas9 sgRNA coverage

KM-12 colorectal carcinoma cancer cell lines were CRISPR-Cas9 screened with Project Score library similarly to Behan et al. [13] using $\times 25$, $\times 50$, $\times 75$, $\times 100$ and $\times 500$ library coverage (Additional file 6: Table S5). Transduction efficiency of KM-12 was maintained at $\sim 30\%$ while cell numbers were adjusted to achieve different levels of library coverage. The different library coverage levels were performed in two independent experiments in technical triplicate; experiment A tested $\times 100$ and $\times 500$ coverage and experiment B tested $\times 25$, $\times 50$, $\times 75$ and $\times 100$.

### Drug perturbed CRISPR-Cas9 screens

We conducted time-series CRISPR-Cas9 screens, performed similarly to Behan et al. [13] in technical triplicate, with dabrafenib treatment in HT-29 cancer cell lines (Additional file 7: Table S6). HT-29 cells were transduced at 30% efficiency on day 1. Following puromycin selection, DNA was extracted on day 8 from a subset of cells representing the baseline undrugged condition. The remaining cells were treated with either dabrafenib (0.1 μM) or DMSO on day 8. Subsequently, DNA extraction, sgRNA amplification and sequencing was performed on day 10, 14, 18 and 21. Read count matrices were processed as described before and statistical analysis to identify the most significantly differential essential genes over-time was performed using R package limma [47] using the F-statistic and respective aggregated $p$ value. $P$ values were adjusted for false discovery rates (FDR) using Benjamini-Hochberg false discovery rate methods. Identical analysis was performed for the original Project Score library and for the in silico down-sample minimal library and then compared.

### Organoid genome-wide CRISPR-Cas9 screens

Genome-wide CRISPR-Cas9 screens were performed in 3 organoids, 1 derived from colorectal carcinoma patient sample (COLO021, ATCC identifier HCM-SANG-0270-C20) and 2 organoids derived from oesophageal cancer (CAM277 and CAM338) (Additional file 8: Table S7). CAM338 was screened in technical duplicate. Organoids were derived and maintained as previously described [48]. To express Cas9, tumoral organoids were dissociated into single cells and incubated overnight in suspension and complete media supplemented with pKLV2-EF1a-BsdCas9-W lentiviral particles and polybrene (8 μg ml–1). The day after, cells were seeded in matrigel and grown as organoids. Blasticidin selection (20 mg/ml) commenced 48 h after transduction and maintained until the end of the experiment. All the organoid lines displayed Cas9 activity over 75%. The genome-wide sgRNA library transduction was adapted from a previous protocol recently reported to screen cancer cell lines [13]. Briefly, tumour organoids were dissociated into single cells and a total of $3.3 \times 10^7$ cells were transduced overnight, in suspension, with an appropriate volume of the lentiviral-packaged whole-genome sgRNA library to achieve 30% transduction efficiency ($\times 100$ library coverage) and polybrene (8 μg ml–1). The following day, cells were seeded in Matrigel and grown as organoids. After 48 h organoids were selected with puromycin (2 mg/ml). After 14 days, approximately $2 \times 10^7$ cells were collected as pellets and stored at $-80\,°C$ for DNA extraction. Genomic DNA was extracted using the Qiagen, Blood & Cell Culture DNA Maxi Kit, 13362 as per the manufacturer's instructions. PCR amplification,

Illumina sequencing (19-bp single-end sequencing with custom primers on the HiSeq2000 v.4 platform) and sgRNA counting were performed as described previously.

### Design of minimal genome-wide CRISPR-Cas9 library

A minimal genome-wide library was assembled from the master reference library by ranking sgRNAs that minimise off-target and maximise on-target effects. Of the 354, 715 guides, 497 did not match any position in GRCh38 and/or targeted any gene and thereby were removed. Additionally, 738 sgRNAs with conflicting gene-targeting annotation across different libraries were also discarded.

Three different groups of sgRNAs corresponding to increasingly relaxed selection stringency levels were defined, termed as green, amber and red. Green represents guides with a single perfect match to the GRCh38 build and no other alignment with one sequence mismatch. Additionally, green sgRNAs have either a JACKS scores within a range between 0 and 2 (Project Score or Avana guides) or a Rule Set 2 score higher than 0.4 (Brunello guides), with the exception to TKOv3 guides where no filter was applied. Amber represents sgRNAs with more relaxed off-target constraints, only requiring a single perfect alignment to the genome, and no filter based on JACKS or Rule Set 2 metrics was used. Lastly, red level sgRNAs can have up to 3 perfect alignments, similar to Koike-Yusa et al. [16], with no filter based on guide efficacy metric, similar to amber sgRNAs.

For all protein-coding genes defined in HGNC [49], we tried to identify 2 optimal sgRNAs within these three different stringency levels. For each gene, guides were ranked using either KS or Rule Set 2 scores, and selection was performed until 2 sgRNAs successfully passed the defined thresholds: (i) the Project Score library was queried and the top 2 sgRNAs ranked by KS scores were picked; (ii) the Avana library was ranked by KS scores and searched to pick the outstanding number of sgRNAs; (iii) the Brunello library was used to pick the outstanding number of sgRNAs and Rule Set 2 scores were used to rank the guides; and lastly (iv), sgRNAs from the TKOv3 library were considered. To minimise library-specific biases, we prioritised the use of sgRNAs originating from the Project Score library.

The assembled minimal library covers 18,761 protein-coding genes with 37,522 sgRNAs (33,986 Project Score; 1732 Brunello; 1493 Avana and 311 TKOv3) with 36,337 green, 740 amber and 445 red confidence level sgRNAs. An additional set of 200 non-targeting sgRNAs, chosen by their similarity to the median fold-changes of all non-targeting guides and with no perfect alignment, no 1nt-mismatch alignment and at most three 2nt-mismatch alignments to the GRCh38 build were added to allow future benchmarks and design improvement. For 107 genes, the sgRNA selection was forced to exclude Project Score library as these generated conflicting gene-level fold-changes (i.e. significant gene essentiality profiles discordant in more than 100 cell lines between the original and minimal library) (Additional file 1: Figure S4c, S4f and S4g).

### Plasmid construction for minimal genome-wide CRISPR-Cas9 library

All plasmids (including the Human sgRNA MinLibCas9) are in the process of being deposited with Addgene. MinLibCas9 sgRNA sequences can be found in Supplementary Table 3 (Additional file 4: Table S3) and primer sequences used in the construction of

the MinLibCas9 sgRNA library can be found in Supplementary Table 9 (Additional file 10: Table S9).

The Human sgRNA MinLibCas9 was constructed as previously described [8]. With the following exception, a modified version of the vector backbone pKLV2-U6gRNA(BbsI)-PGKpuro2ABFP-W [8] was used to improve downstream pooled oligonucleotide cloning. Briefly, the ccdB resistance gene cassette was PCR amplified from pKLV1-fl-U6gRNA(BbsI)-ccdB-PGKpuro2ABFP (a kindly provided by E. Metzokapian) with the following oligos, Gibson_pKLV2-ccdb_Fwd and Gibson_pKLV2-ccdb_Rev, which included appended oligonucleotide sequences complementary to the pKLV2-U6gRNA(BbsI)-PGKpuro2ABFP-W backbone and BbsI restriction enzyme sites. The amplified ccdB gene product was cloned into the pKLV2-U6gRNA(BbsI)-PGKpuro2ABFP-W plasmid, linearized by BbsI digestion, by Gibson Assembly Cloning (New England Biolabs) to generate the modified pKLV2-U6gRNA(BbsI)-ccdB-PGKpuro2ABFP-W plasmid vector. This was subsequently transformed into ccdB Survival *E. coli* (Invitrogen).

A single-stranded oligonucleotide pool containing all guide RNA sequences was synthesised by TWIST Bioscience. Sequenced oligonucleotides included primer appends for generating double-stranded oligonucleotides compatible for Gibson assembly into the pKLV2-U6gRNA(BbsI)-ccdB-PGKpuro2ABFP-W plasmid vector (i.e. Pool_PCR_Fwd and Pool_PCR_Rev). Pools were amplified using Q5 Hot Start High-Fidelity 2X Master Mix (New England Biolabs), PCR cycling conditions: 98 °C for 30s, 98 °C for 10s, 67 °C for 10s and 72 °C for 15 s, for 14 cycles with a final extension of 72 °C for 2 min. Amplicons were PCR-purified using (DNA clean and concentrator kit, Zymo) and cloned into 100 ng of BbsI linearized pKLV2-U6gRNA(BbsI)-ccdB-PGKpuro2ABFP-W vector by Gibson Assembly Cloning (New England Biolabs) following the manufacturer's instructions. Multiple Gibson assemblies were pooled, ethanol precipitated and transformed into 200 l of electrocompetent *E. coli* (Lucigen Endura™ ElectroCompetent Cells, Lucigen). 4 × 250 ml LB preps (supplemented with 100 µg mL−1 carbenicillin) were inoculated and grown at 37 °C for 16 h. Plasmid DNA (pDNA) was extracted using a Qiagen maxi prep kit.

Illumina sequencing of sgRNAs and analysis of guide distribution was performed as follows. PCR amplification, Illumina sequencing (19-bp single-end sequencing with custom primers on the HiSeq2000 v.4 platform) and sgRNA counting were performed as described previously. For the plasmid library, total read counts per sgRNA were calculated using the count_spacers python script [50]. Illumina sequencing of the plasmid library identified 88.3% of reads were perfect matches when compared to predicted sgRNA sequences. All guides were detected. The read distribution of each sgRNA was determined using the Ineq package in R (version 3.5.3) to calculate both the Lorenz-curve and Gini-coefficient (Additional file 1: Figure S8a and S8b).

Lentiviral vectors and packaging plasmids (psPax2 and pMD2.G, Addgene) were transfected into 60% confluent HEK293 cells using Lipofectamine LTX (Gibco) at the following ratio: 7.5 µg lentiviral vector (MinLibCas9 plasmid DNA), 18.5 µg psPax2 and 4 µg pMD2.G per 15 cm dish.

### CRISPR-Cas9 MinLibCas9 HT-29 screens
Titrations of the lentiviral packaged sgRNA library were performed to identify the volume of supernatant to achieve 30% transduction efficiency, cells were analysed for

stable BFP expression using flow cytometry 72 h post transduction. Using this volume viral transduction of $12.5 \times 10^6$ ($\times 100$ coverage of the sgRNA library, consistent with what was used for the original Project Score dataset) Cas9 expressing HT-29 cells was performed in technical triplicate. Seventy-two hours post viral transduction, 30% transduction efficiency was confirmed by flow cytometry and 1 week selection with 2 μg/ml puromycin started. At 14 days post library transduction cells were harvested and $2.5 \times 10^7$ cells were pelleted; library coverage was also tested by analysis of BFP expression and found to be 62%. A minimum of $1.9 \times 10^7$ cells were maintained throughout the 2-week screen. Genomic DNA extraction and amplification was performed as per Behan et al. [13], and sequencing was performed on a HiSeq2500.

### Project Score vs MiniLibCas9 BAGEL outcomes comparison

We selected MinLibCas9 library guides that are included in the Project Score library and used them to compute gene-level Bayesian Factors (BFs) with BAGEL (Hart and Moffat 2016). We then computed a Pearson correlation coefficient for each cell line comparing the Project Score BFs and the MinLibCas9 BFs considering only shared genes between the two libraries. Next, Project Score essential and nonessential genes (at 5% FDR) from Project Score were derived for each cell line and used as positive/negative sets to compute the area under the precision-recall curve (AUPRC) obtained considering the MiniLibCas9 BFs as a rank classifier.

### Code availability

All code and results are publically available and distributed are distributed under the open-source 3-Clause BSD License at https://github.com/EmanuelGoncalves/crispy/tree/master/notebooks/minlib [51] and https://doi.org/10.5281/zenodo.4313863 [52].

### Supplementary Information

**Additional file 1: Figure S1.** Randomised selection of n sgRNAs per gene. ROC curve derived from previously defined sets of essential and non-essential genes [6]. Standardized partial area under the ROC curve (AROC) calculated per cell line over the range of maximum false discovery rate of 20%. AROCs are compared between downsampled sgRNAs and all sgRNAs available for all the covered genes. 10 random sgRNAs permutations without replacement per cell line and per n guides were performed and AROCs mean values are plotted. AROC were calculated for each replicate of Project Score [13] ($n$ = 663). **Figure S2.** MinLibCas9 sgRNA rank and selection flowchart and sgRNA metrics comparison. a, flowchart describing the sgRNA selection procedure. b, efficiency metrics of Project Score library sgRNAs - KS metric (Kolmogorov Smirnov test) comparison to non-targeting guides, JACKS scores [14], Rule Set 2 [9] scores, and FORECasT [21] predicted percentage of in frame deletions produced - plotted together with guides median fold-changes calculated across 663 samples. Spearman correlation coefficients are reported in the lower triangle of the grid. Plots in the diagonal represent the distribution of the respective metric. **Figure S3.** Down-sample analysis of top n sgRNAs ranked using KS and JACKS metrics. Combined score discards sgRNAs with a JACKS score outside the range of [0, 2] and then selects the top n sgRNAs according to the KS score (descending order, stronger KS scores and thereby stronger absolute fold-changes). Essential/Non-essential AROCs are the area under the ROC curve (at 20%FDR) using known essential and non-essential genes. Precision and recall rates are calculated between the sets of significant gene-level dependencies (at 1% FDR) estimated using the original library and the down-sampled library. Each box-and-whisker plot shows 1.5 x interquartile ranges and 5–95th percentiles, centres indicate medians ($n$ = 245 cancer cell lines from Project Score [13]). **Figure S4.** MinLibCas9 benchmarking. a, distributions of CRISPOR sgRNA scores. b,distributions of KS, JACKS and RuleSet2 scores for sgRNAs from Project Score library that contain T-stretches of 4 and 5. c, significant dependencies identified using different FDR rates between essential and nonessential genes (1%, 5%, 10%, 15%, 20% and 25%). MinLibCas9 fold-change thresholds at each FDR rate and the average precision (AP) of the gene dependencies identified in MinLibCas9 that are also found with Project Score library. Error bars represent standard deviation. d, average precision scores per cell line of established dependencies with MinLibCas9 fold-changes, calculated through downsampling the original screens, correlated with Project Score data quality as measured by the area under the ROC curve (AROC) using known essential and non essential genes. e,MinLibCas9 fold-change threshold for each cell line identified at 1% FDR of essential versus non-essential genes plotted against those identified with Project Score library .

Colour represents density. f, cumulative number of dependencies (at 1% FDR) identified in 245 cancer cell lines for each gene with both the full original library and the minimal library. g,scaled fold-changes (median essential genes fold-change = 1; median non-essential genes fold-change = 0) of dependencies recapitulated (*n* = 251,387) and missed (*n* = 36,996) with the minimal library (two-sided Welch's t-test *p*-value < 0.001). Each box-and-whisker plot shows 1.5 x interquartile ranges and 5–95th percentiles, centres indicate medians. **Figure S5.** sgRNA library coverage analysis in KM-12 cancer cells. a, AROC of essential/non-essential genes at different guide coverage levels. b, technical replicates correlation. Comparisons are made between the original Project Score library and the in silico minimal library . c and d, show correlation plots between the different library coverage for Project Score and downsampled MinLibCas9 libraries, respectively. A and B represent two independent experiments. **Figure S6.** CRISPR-Cas9 dropout screens upon treatment with Dabrafenib. a, diagram of the experimental setup. b, gene fold-changes averaged across the different time points (day 8, 10, 14, 18 and 21) obtained using Project Score library compared to the in silico MinLibCas9. Colour represents point density. c, time-series fold-changes of the top significantly essential hits (compared to control experimental arm, DMSO) obtained with Project Score and MinLibCas9 library, the values of three technical replicates are represented with the mean and by the minimum and maximum values as error bars. **Figure S7.** CRISPR-Cas9 loss-of-fitness screens in 3D organoids. a, Comparison of gene fold-changes obtained using the in silico minimal and original library in a colon carcinoma organoid (COLO021) and two oesophageal adenocarcinoma organoids (CAM277 and CAM338). b, AROC of essential/non-essential genes of each organoids obtained with both libraries. c, CAM338 technical replicates correlation. Colour coding represents point density. **Figure S8.** MinLibCas9 screens in HT-29 cancer cell line. a, Distribution of guides within MinLibCas9 library plasmid calculated to be comparatively even. Skew ratio of top 10% vs bottom 10% guides = 1.75 (minimum read count = 30, maximum read count 7698). b, lorenz curve of the MinLibCas9 plasmid library, Gini coefficient = 0.12. c, correlation of gene level fold-changes of the three technical replicates of MinLibCas9 screens versus the three technical replicates of HT-29 screens performed in Project Score. Spearman's Rho correlations are reported on the lower diagonal, fold-changes distributions are displayed in the diagonal, and upper diagonal shows the scatter plots between the two samples fold-changes colored by density. d, recall of previously defined essential (upper panel) and non-essential (lower panel) genes quantified by the area under the recall curve. e,distributions of gene level fold-changes for each sample. Box-and-whisker plot shows 1.5 x interquartile ranges and 5–95th percentiles, centres indicate medians. **Figure S9.** Application of common CRISPR-Cas9 analytical pipelines to MinLibCas9. a,sgRNA fold-change distributions of HT-29 MinLibCas9 screens before (Original) and after copy number bias correction using supervised (Crispy [28]) and unsupervised (CRISPRcleanR [34]) approaches across the different levels of copy number amplifications. b, distribution of the correlation coefficients between the downsample MinLibCas9 and Project Score BAGEL bayesian factors across Project Score CRISPR-Cas9 screens (*n* = 325). c, average precision (AP) scores of the precision-recall curves of BAGEL significantly essential genes from Project Score (at 5% FDR) obtained considering MinLibCas9 BAGEL bayesian factors as rank classifier (one point per cell line). Vertical line represents the percentage of cell lines with at least 80% AP score. d, distributions of the BAGEL bayesian scores calculated with the Project Score for the genes found to be non-essential with both Project Score and downsample MinLibCas9 (Agree Non-essential), essential with both libraries (Agree Essential), and that are essential in only one library (Disagree).

**Additional file 2: Table S1.** Median number of sgRNAs per gene and library size of currently available human genome-wide CRISPR-Cas9 libraries.

**Additional file 3: Table S2.** Reference CRISPR-Cas9 library containing sgRNAs originating from multiple libraries with standardised genomic annotation and guide efficiency metrics.

**Additional file 4: Table S3.** Genome-wide minimal human CRISPR-Cas9 library, MinlibCas9.

**Additional file 5: Table S4.** KS scores estimated for sgRNAs of Project Score and Avana libraries.

**Additional file 6: Table S5.** Raw counts of the CRISPR-Cas9 screens at different guide coverage performed in KM-12 cancer cell line.

**Additional file 7: Table S6.** sgRNA counts of the CRISPR-Cas9 screens followed by drug treatment with dabrafenib in HT-29 cells.

**Additional file 8: Table S7.** CRISPR-Cas9 raw counts for three different organoids derived from cancer samples.

**Additional file 9: Table S8.** MinLibCas9 raw counts for three technical replicates of HT-29.

**Additional file 10: Table S9.** Oligonucleotide and primer sequences.

**Additional file 11.** Review history.

#### Acknowledgements
We acknowledge Joel Rein for helpful comments on the integration of WGE annotations. We would also like to thank the core support provided by the Cellular Operations teams, namely Cellular Generation and Phenotyping (CGaP), Gene Editing (GE), Flow Cytometry (CCF) and Cellular RnD, with specific thanks to Lucy Kitchin and Claire Hardy.

#### Peer review information
Yixin Yao and Kevin Pang were the primary editors on this article and managed its editorial process and peer review in collaboration with the rest of the editorial team.

#### Review history
The review history is available as Additional file 11.

#### Authors' contributions
Conceptualization: E.G., M.G. Software: E.G., M.T., C.P., F.A. Validation: F.B., G.P., M.S., D. A. J., C. M. B., S. P., O.D. Formal analysis: E.G., A.V. Data curation: E.G., M.T. Writing - original draft preparation: E.G., M.G. Writing - reviewing and editing:

All authors. Visualisation: E.G. Supervision: O.D., D.P.S., F.I., L.P., K.Y., M.G. Funding acquisition: M.G. The authors read and approved the final manuscript.

### Funding
Work in M.J.G was funded by the Wellcome Trust (206194) and Open Targets. L.P. is supported by Wellcome (206194), Estonian Research Council (IUT34-4), and Estonian Centre of Excellence in IT (EXCITE) (TK148).

### Availability of data and materials
Analysis code is available in the Github repository (https://github.com/EmanuelGoncalves/crispy/tree/master/notebooks/minlib, doi:https://doi.org/10.5281/zenodo.4313863), released under 3-Clause BSD License [51, 52]. The Sanger Project Score CRISPR-Cas9 datasets raw counts were downloaded from https://score.depmap.sanger.ac.uk/. The Broad DepMap19Q2 public CRISPR-Cas9 datasets were downloaded through the portal https://depmap.org/portal/ and in the figshare repository, https://figshare.com/articles/DepMap_19Q2_Public/8061398/1. All data generated for this study are contained in supplementary information.

### Ethics approval and consent to participate
Not applicable.

### Competing interests
M.G. has performed consultancy for Sanofi, receives research funding from AstraZeneca and GSK, and is co-founder of Mosaic Therapeutics. F.I. receives funding from Open Targets, a public-private initiative involving academia and industry, and performs consultancy for the joint CRUK - AstraZeneca Functional Genomics Centre. L.P. receives funding from Open Targets. All other authors declare no competing interests.

### Author details
[1]Wellcome Sanger Institute, Wellcome Genome Campus, Hinxton, UK. [2]Open Targets, Wellcome Genome Campus, Hinxton, Cambridge CB10 1SA, UK. [3]Human Technopole, Via Cristina Belgioioso 147, 20157 Milan, Italy. [4]Department of Computer Science, University of Tartu, 18 Narva St, Tartu, Estonia. [5]Institute for Frontier Life and Medical Sciences, Kyoto University, Kyoto 606-8507, Japan.

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

## 