## [**Additional file 11.** Review history. · Genome Biology]

Review History

First round of review

Reviewer 1

Are you able to assess all statistics in the manuscript, including the appropriateness of statistical tests used? Yes, and I have assessed the statistics in my report.

Comments to author:

Goncalves and colleagues reported a workflow to evaluate, rank and assemble a minimal genome-wide human CRISPR-Cas9 library, which they named as 'MinLibCas9', from over 300,000 unique sgRNAs from multiple broadly adopted libraries. This optimized minimal library consists 2 sgRNAs for each targeting gene, which significantly reduced the typical library size that was required to knockout all human genes, but preserved assay sensitivity and specificity. Additionally, in order to benchmark the MinLibCas9 library, the authors provided three cases to demonstrate the library performance. There're two selling points from this manuscript, one is the minimal and optimized MinLibCas9, and the other is a data-driven approach to prioritize the most effective sgRNAs. However, both points lack sufficient evidences. Additionally, the contents of the manuscript weren't well organized, which makes it a bit difficult to follow.

Major points:

1. Multiple metrics from available tool sets were incorporated into the "reference library" and were used to rank the sgRNAs. Based on these metrics, a more comprehensive quality control of the MinLibCas9 should be provided, e.g. isoform coverage, functional domain coverage, number of hits on the genome etc.
2. The author used KS score to select sgRNAs, however, this score is weakly correlated with the predicted on-target efficiency (supplementary fig 2). Intuitively, for essential genes, the correlation should be reasonably well, since "good" sgRNAs should have high on-target efficiency in general. For non-essential genes, they should also be correlated to some extent since high rate of DNA DSB will be introduced by genes with high on-target efficiency. It raised a big concern that whether the KS score is a good indicator for sgRNA performance.
3. The KS score from loss-of-fitness screen was used to prioritize the sgRNAs. And the selected sgRNAs performed well in a loss-of-fitness benchmarking (Fig2b). However, in another benchmarking using positive selection screening, the Spearman correlation quickly dropped to 0.72 (supplementary Fig6), as well as in the validation of 3D organoid culture (supplementary Fig7). It brought up another concern that whether the MinLibCas9 was overfit to the loss-of-fitness screening but not in other types of screening experiments.
4. Relevant to the previous point, an independent benchmarking should be performed by using the MinLibCas9 and a well-known library (e.g. Brunello) in parallel to demonstrate the performance. As a subset of the Project Score, the similar performance of the MinLibCas9 and the Project Score can only support the idea that the selective sgRNA can well represent the original large-size library in loss-of-fitness screening, but not sufficiently support that the MinLibCas9 is an independent high quality library.
5. One of the two major points is the data-driven approach for sgRNA selection. However, this part of efforts was poorly described through the manuscript. In many cases, the authors stated what they have done. However, as a methodology paper, it is important to let the audience understand what the rationale is behind. Also, at least one flowchart should be included to clarify the entire computational procedure.

Minor points:

1. It is not clear which dataset was used in different parts of the manuscript. For example, how many datasets have been used in the supplementary figure 1? Either the main text or the figure legend did not provide the information.

2. Multiple available guide efficacy metrics were used to evaluate the on-target efficacy, but it is not clear how these metrics were picked. There're three major categories of algorithms to evaluate guide efficacy, from nucleotide composition, using training model and using deep learning algorithm. It will be more convincing to include at least one representative metric from each method.
3. Whether KS scores were standardized for sgRNAs selected from multiple datasets?
4. In Fig2b, why the precision score of a few cell lines are very low compared to most of others?
5. The library should be deposited to public domain as other well-known libraries for easy access.

Reviewer 2

Are you able to assess all statistics in the manuscript, including the appropriateness of statistical tests used? Yes, and I have assessed the statistics in my report.

Comments to author:

Goncalves et al describe a genome-scale CRISPR/Cas9 library that is roughly half the size of current libraries, presenting a major step forward in CRISPR screen miniaturization that will substantially reduce the time, effort, and expense of CRISPR screening while (in large part) maintaining some degree of backwards-compatibility with the extant body of CRISPR screen data to facilitate integrated data analysis. This latter point is not stressed enough in the study, as it is a key differentiator between the described 38k Cas9 library and the previously described 17k Cas12a/Cpf1 library described in Liu et al, Nat Comms 2019. That this paper is not cited is a serious oversight on the part of the authors, and its inclusion in Figure 1 would significantly alter the framing of this paper.

Overall, this manuscript is a case study in how to design and validate a new library. The comparisons are robust, the experimental validations appropriate. The authors note the relative ease of identifying "good" Cas9 guides targeting essential genes but the difficulty of doing so when targeting the 2/3 of the genome that does not show a knockout phenotype. The approach they use, identifying consistent phenotype with a combination of JACKS scores and comparison with non-targeting gRNA, is inspired and should be effective.

The authors are also to be commended on their exhaustive supplementary data, including raw data in supplementary tables.

Two additional questions should be addressed before publication (in addition to the point noted in the first paragraph):

1. What are the implications for analysis of a screen performed using this library? The Bayesian approach of JACKS will work best with more data, not less. It is noted that the authors use mean fold change in their analysis in Supp Fig 6. Good AUPR separating gold-standard essentials vs. nonessentials is critical and well-supported but, if the task is finding differential essentials, what does it mean to have half as much data?
2. Similarly, what does the reduction in genome coverage mean for normalization techniques like CERES or CRISPRcleanR? Is there reduced ability to identify and correct for segment amplicions, increasing the likelihood for false positives?

Minor points:

Figure 2: typo in Fig2b y-axis label ('precession').

Supp Fig 2: are these correct? It is difficult to believe that RuleSet2 vs Median is basically uncorrelated ($R=-0.03$), while FORECasT vs Median is near-perfect (anti)correlation ($R=-0.97$).

Supp Fig 5: what are the A/B annotations in the X axis labels?

Supp Fig 7: excellent use of RMSE in scatter plots designed to show correlation. This is how everyone should do it.

Reviewer 3

Are you able to assess all statistics in the manuscript, including the appropriateness of statistical tests used? Yes, and I have assessed the statistics in my report.

Comments to author:

Goncalves et al. present a minimal genome-scale CRISPR-Cas9 library with 2 sgRNAs targeting each gene. This approach will decrease costs and enable many applications. This represents a valuable resource for the functional genomics community.

Major concerns:

The key question that the paper must address in as many ways as possible is: how can the authors ensure that by selecting for non-essential guides with the highest DNA cutting toxicity that they are not somehow selecting for unpredicted off-target activity? Is all of this analysis DNA copy number corrected? DNA copy number amplification increases the Cas9 DSB cutting effect at both on and off target sites. The authors try to examine this by looking at the deviation between guide phenotypes for guides that target the same gene but given that the original Cas9 sgRNA libraries that they are using to analyze the data only have 4-10 sgRNAs they are working with limited information on the distribution of phenotypes expected for guides with high on or off target activity which limits how robust this outlier analysis can be.

In this paper the authors have expected genes that should be identified but in general this is not the case in functional genomics screens that are not based on identification of essential genes (e.g. genes that modulate response to a new drug that hasn't previously been tested with a larger library). With just two sgRNAs per gene the authors should provide guidance on how they expect to call hit genes in functional genomics assays. Are they expecting people will only use the log2 fold change or Z scores relative to the negative control distribution? With only two guides most statistical tests can't be used. Ideally the authors can use existing or new data to provide an example of how to do this and how robust this can be.

Please show replicate scatter plots rather than just pearson summary statistics for Figure S5B as in Figure S8C. It is quite surprising that the authors claim coverage as low as 25x is sufficient for a genome scale screen. It had previously been shown/claimed that higher coverage in such assays is correlated with increased reproducibility (<https://www.nature.com/articles/nrg.2017.97>). Why is this data only shown for KM-12 cells? Why did the authors switch to using a pearson correlation in Figure S5B but everywhere else use a spearman? What coverage was Project Score run at? Are the authors really saying they think functional genomics screens should be run at say 50x coverage?

This manuscript is only useful/impactful to other researchers in the field if the sgRNA library is made public through addgene or some other open access source.

Response to reviewers

We thank the reviewers for their constructive feedback on our manuscript and believe we have been able to address their comments. Below follows our point-by-point reply to the reviewers' comments (provided in italics), and our responses as well as significant changes to the manuscript text are highlighted in blue font.

“““

Reviewer #1: Goncalves and colleagues reported a workflow to evaluate, rank and assemble a minimal genome-wide human CRISPR-Cas9 library, which they named as 'MinLibCas9', from over 300,000 unique sgRNAs from multiple broadly adopted libraries. This optimized minimal library consists 2 sgRNAs for each targeting gene, which significantly reduced the typical library size that was required to knockout all human genes, but preserved assay sensitivity and specificity. Additionally, in order to benchmark the MinLibCas9 library, the authors provided three cases to demonstrate the library performance.

There're two selling points from this manuscript, one is the minimal and optimized MinLibCas9, and the other is a data-driven approach to prioritize the most effective sgRNAs. However, both points lack sufficient evidences. Additionally, the contents of the manuscript weren't well organized, which makes it a bit difficult to follow.

Major points:

1. Multiple metrics from available tool sets were incorporated into the "reference library" and were used to rank the sgRNAs. Based on these metrics, a more comprehensive quality control of the MinLibCas9 should be provided, e.g. isoform coverage, functional domain coverage, number of hits on the genome etc.

”””

We agree that regulatory and functional information, such as protein domains [1,2], could represent valuable information for the design and interpretation of the library reagents. We have improved the metrics provided for the reference/MinLibCas9 library, with the addition of PFAM identifiers and names of functional domains that overlap the cut-site of each gRNA. Based on this data, our MinLibCas9 library contains 5% more gRNAs that overlap a known PFAM domain versus the reference library. Although not explicitly stated, the “number of hits

on the genome“ for each gRNA was already available within our dataset, and is defined by the 0 mismatch for the Off_Target values in SupplementaryTable2.

With regards to isoform coverage, we are concerned that this metric would be too subjective to enable consistent scoring of the gRNAs. Each of the original libraries were designed using gene annotation from either the RefSeq or Ensembl resources, which use different methods to annotate transcripts. The current MANE project aims to resolve inconsistencies between these datasets and provide a minimal set of matching annotation, however it does not include coverage for every gene.

“““

2. The author used KS score to select sgRNAs, however, this score is weakly correlated with the predicted on-target efficiency (supplementary fig 2). Intuitively, for essential genes, the correlation should be reasonably well, since "good" sgRNAs should have high on-target efficiency in general. For non-essential genes, they should also be correlated to some extent since high rate of DNA DSB will be introduced by genes with high on-target efficiency. It raised a big concern that whether the KS score is a good indicator for sgRNA performance.

”””

We agree with the reviewer that sgRNAs targeting essential genes, compared to non-essential, should carry stronger negative effects and therefore have strong associations across different metrics. Consistently, sgRNA metrics' correlations are stronger when considering essential genes targeting guides only (Rebuttal Figure 1a) versus non-essential genes targeting guides only (Rebuttal Figure 1b). Nonetheless, correlation patterns are mostly kept.

Rebuttal Figure 1. sgRNA efficiency metrics correlation. a, correlations using essential and b, using non-essential genes.

This is not unexpected because we can not confidently say that current sgRNA efficiency metrics represent a ground truth of sgRNA on-target efficiency. Supporting this, the lack of agreement between different metrics has been reported and discussed before [3,4]. Nonetheless, we believe this emphasises the different nature of each metric, for example Rule Set 2 scores capture sequence-based information, while JACKS is an empirical score that compares sgRNAs targeting the same gene. Hence, rather than correlating strongly we believe these metrics contain complementary sgRNA information that together can be used to better select and rank sgRNAs than each metric independently. In Supplementary Figure 3, we show that JACKS and KS scores combined provide improved identification of essential genes and recall of previously identified cell line dependencies.

““““

3. The KS score from loss-of-fitness screen was used to prioritize the sgRNAs. And the selected sgRNAs performed well in a loss-of-fitness benchmarking (Fig2b). However, in another benchmarking using positive selection screening, the Spearman correlation quickly dropped to 0.72 (supplementary Fig6), as well as in the validation of 3D organoid culture (supplementary Fig7). It brought up another concern that whether the MinLibCas9 was overfit to the loss-of-fitness screening but not in other types of screening experiments.

””””

The reviewer raises an important point. Firstly, Figure 2b reports average precision scores which indeed range above 90% for most cell lines, although the supplementary figures the

reviewer refers to (Supplementary Figure 6 and Supplementary Figure 7) both report a different metric and assessment, i.e. fold-change spearman correlations, therefore these are not directly comparable. In fact, fold-change correlations between MinLibCas9 and Project Score libraries are very consistent across different analysis: (i) HT-29 MinLibCas9 screens (Spearman's Rho 0.62, Figure 2d); (ii) downsample positive selection (Spearman's Rho 0.72, Supplementary Figure 6b); and (iii) downsample 3D organoid screens (Spearman's Rho 0.69 - 0.71, Supplementary Figure 7a).

Secondly, Project Score library has been extensively benchmarked before and has shown good agreement with other independent libraries [3,5–8], as is the case of all the other libraries considered to assemble the reference master library. The fact that 90.6% of the MinLibCas9 guides originate from Project Score library, which we agree it could be seen as overfitting, has the very important benefit of ensuring backward compatibility to a high quality library, that has been extensively validated, and which was used to perform hundreds of screens already. This is very important as even large libraries, >4 sgRNAs per gene, generate batch effects that need to be analytically corrected prior to full integration with other libraries [6,8–10]. This requires that a large enough sample size is available with the smaller library to make this normalisation possible. Hence, ensuring high similarity to one library is important for MinLibCas9 as its most relevant application is to screen hard to growth or complex models which inevitably will be performed in low numbers.

Thirdly, we expanded the comparison to other libraries and considered A375 and HT-29 cancer cells which, between them, have been screened with the Avana, Brunello, Project Score and MinLibCas9 (new screening data, not a down-sample) libraries. Considering most genes display little variation we focused on a previous defined set of Strongly Selective Dependencies (SSD) genes [6]. Reassuringly all HT-29 or A375 samples clustered together irrespective of the library (Rebuttal Figure 2). MinLibCas9 HT-29 samples have stronger similarity to Project Score HT-29 samples, although these are also very similar to Avana's HT-29 screens. Overall, this suggests that MinLibCas9 generates results similar to other commonly used and validated CRISPR-Cas9 libraries, as it would be expected considering that Project Score performs similarly.

Rebuttal Figure 2. HT-29 and A375 screens correlation. Strongly Selective Dependencies (SSD) [6] genes fold-change Spearman's correlation across CRISPR-Cas9 screens performed with Avana, Brunello, Project Score and MinLibCas9 in HT-29 and A375 cancer cell lines.

““““

4. Relevant to the previous point, an independent benchmarking should be performed by using the MinLibCas9 and a well-known library (e.g. Brunello) in parallel to demonstrate the performance. As a subset of the Project Score, the similar performance of the MinLibCas9 and the Project Score can only support the idea that the selective sgRNA can well represent the original large-size library in loss-of-fitness screening, but not sufficiently support that the MinLibCas9 is an independent high quality library.

””””

We have addressed this reviewer's point in conjunction with the previous major suggestion, please refer to our previous reply.

““““

5. One of the two major points is the data-driven approach for sgRNA selection. However, this part of efforts was poorly described through the manuscript. In many cases, the authors

stated what they have done. However, as a methodology paper, it is important to let the audience understand what the rationale is behind. Also, at least one flowchart should be included to clarify the entire computational procedure.

””””

We apologise for the lack of clarity. We have improved the text and added a flowchart to Supplementary Figure 2a detailing the selection of sgRNAs for MinLibCas9 from the reference master library.

“”“

Minor points:

1. It is not clear which dataset was used in different parts of the manuscript. For example, how many datasets have been used in the supplementary figure 1? Either the main text or the figure legend did not provide the information.

””””

We agree and added a description specifying the data-sets used in each analysis.

“”“

2. Multiple available guide efficacy metrics were used to evaluate the on-target efficacy, but it is not clear how these metrics were picked. There're three major categories of algorithms to evaluate guide efficacy, from nucleotide composition, using training model and using deep learning algorithm. It will be more convincing to include at least one representative metric from each method.

””””

We agree and appreciate that other metrics exist and these might represent potential improvements to our selection procedure. The main limitation to do so is that systematic calculation and integration of other metrics requires that existing tools can be queried in a systematic way so that we can annotate all 300,167 sgRNAs from the Master Reference library. Nonetheless, we envisioned integrating as many metrics as possible, i.e. JACKS, FORECasT, Rule Set 2, MIT specificity and CrisprScan. While we could envision using all metrics as part of the sgRNA selection and ranking we sought to leave out metrics for independent validation, i.e. MIT specificity and CrisprScan.

“”“

3. Whether KS scores were standardized for sgRNAs selected from multiple datasets?

””””

The Avana and Project Score libraries' KS-scores were calculated and used separately, thus not requiring them to be standardized across multiple data-sets. Nonetheless, it is a good point that it will be important to assess potential effects on KS scores depending on the data-set utilised. The fact that they share a similar scale, 0 to 1, and use a similar set of sgRNAs, i.e. non-targeting, will help mitigate potential batch effects.

“““

4. *In Fig2b, why the precision score of a few cell lines are very low compared to most of others?*

”””

Cell lines average precision (AP) scores are strongly correlated with the AROC of essential/non-essential genes), which is a proxy for screen quality. We added this as a new panel to Supplementary Figure 4d (also for convenience see below as Rebuttal Figure 3). In fact the two lowest AP ranking cell lines, GCIY (SIDM00274) and D-247MG (SIDM00622), perform very poorly in our quality control assessment [7]. Thus, this suggests that cases where the MinLibCas9 does not recapitulate the dependencies found with the Project Score library are likely associated with poor sample quality.

Rebuttal Figure 3. Screens quality relation with average precision of dependencies. Correlation of average precision scores obtained recapitulating established dependencies from Project Score using MinLibCas9 fold-changes calculated through downsampling original screens (Figure 2b).

“““

5. *The library should be deposited to public domain as other well-known libraries for easy access.*

”””

We absolutely agree and we are in the process of depositing it in Addgene.

“““

Reviewer #2: Goncalves et al describe a genome-scale CRISPR/Cas9 library that is roughly half the size of current libraries, presenting a major step forward in CRISPR screen miniaturization that will substantially reduce the time, effort, and expense of CRISPR screening while (in large part) maintaining some degree of backwards-compatibility with the extant body of CRISPR screen data to facilitate integrated data analysis. This latter point is not stressed enough in the study, as it is a key differentiator between the described 38k Cas9 library and the previously described 17k Cas12a/Cpf1 library described in Liu et al, Nat Comms 2019. That this paper is not cited is a serious oversight on the part of the authors, and its inclusion in Figure 1 would significantly alter the framing of this paper.

”””

We have emphasised better the backward compatibility with existing Project Score library and the benefit this represents when integrating with existing large banks of CRISPR-Cas9 screens of cancer cell models. This is now specified in the abstract, introduction and discussion. We apologise for the oversight of not mentioning Liu et al 2019. This is indeed a very relevant publication. Our focus for Figure 1, and the whole study, was systems based on Cas9 only, Liu et al 2019 takes advantage of Cas12a and a multiplexed system. We have changed the discussion to reference this work.

“““

Overall, this manuscript is a case study in how to design and validate a new library. The comparisons are robust, the experimental validations appropriate. The authors note the relative ease of identifying "good" Cas9 guides targeting essential genes but the difficulty of doing so when targeting the 2/3 of the genome that does not show a knockout phenotype. The approach they use, identifying consistent phenotype with a combination of JACKS scores and comparison with non-targeting gRNA, is inspired and should be effective.

The authors are also to be commended on their exhaustive supplementary data, including raw data in supplementary tables.

”””

We are very grateful for the reviewer's positive comments.

“““

Two additional questions should be addressed before publication (in addition to the point noted in the first paragraph):

1. *What are the implications for analysis of a screen performed using this library? The Bayesian approach of JACKS will work best with more data, not less. It is noted that the authors use mean fold change in their analysis in Supp Fig 6. Good AUPR separating gold-standard essentials vs. nonessentials is critical and well-supported but, if the task is finding differential essentials, what does it mean to have half as much data?*

””””

The reviewer is absolutely correct that reducing the number of sgRNAs per gene comes with a challenge of having higher uncertainty on the gene-level measurements, as we report with lower replicate correlations (Supplementary Figure 5), and current methods to analyse genome-wide CRISPR-Cas9 libraries likely need to be adapted to work more robustly with minimal libraries, e.g. Low Fat BAGEL [11]. For sgRNA efficiency metrics, such as JACKS, it will be difficult to have robust estimations considering that only two guides per gene are available. On the other hand, it will still be possible to estimate KS scores with MinLibCas9 library since it also includes a set of 200 non-targeting sgRNAs.

Regarding the identification of gene essentialities, we believe that a 2 sgRNA / gene minimal library can be used to robustly identify differential gene essentialities. Firstly, our downsample analysis suggests that we can find the vast majority of gene essentialities identified with Project Score library (5 sgRNAs / gene) using MinLibCas9 (Figure 2b), and those that are missed are associated with lower quality samples and borderline cases with weaker dependencies (Supplementary Figure 4d and 4f). Additionally, as the reviewer pointed out, in a positive selection experiment utilising a downsample version of MinLibCas9 we successfully recapitulated time-series gene dependency profiles and assessed their statistical significance (Supplementary Figure 6c). Secondly, we set out to test the compatibility of MinLibCas9 with a commonly used method to estimate significant CRISPR-Cas9 dependencies, BAGEL [12]. We downsampled MinLibCas9 from Project Score library and screens (325 cancer cell lines) and observed that: (i) cell line BAGEL bayes scores show a median Pearson's R of 88.7%; (ii) binary gene dependencies (5% BAGEL) found with MinLibCas9 show a strong agreement with the BAGEL bayes factors calculated with Project Score library with a median average precision of 89.3%; and (iii) genes not found essential (at 5% FDR with BAGEL) with both libraries have significantly lower BAGEL bayes scores with Project Score library (Mann-Whitney U statistic p-value < 1e-4), indicating that these are weaker dependencies and therefore borderline calls. We added these results to the manuscript as a new Supplementary Figure 9 b, c and d.

Overall, we think MinLibCas9 can robustly recapitulate differential dependencies and be analysed with commonly used tools and approaches for CRISPR-Cas9 screens, such as BAGEL. Nonetheless, we note that analytical modifications might be warranted to better account for the substantial decrease in reagents and increase in noise.

“““

2. Similarly, what does the reduction in genome coverage mean for normalization techniques like CERES or CRISPRcleanR? Is there reduced ability to identify and correct for segment amplicons, increasing the likelihood for false positives?

”””

The reviewer raises an important point. We have taken the CRISPR-Cas9 screens performed with MinLibCas9 in HT-29 cancer cells, which have several copy number amplifications including in chromosome 8 around the MYC locus, and performed a supervised and unsupervised correction of the copy number biases, using Crispy [13] and CRISPRcleanR [14], respectively (Supplementary Figure 9). Supervised correction takes copy number segments as input. For large copy number amplifications where the bias starts to be noticeable (absolute copy number ≥ 8) both approaches attenuated substantially the mean fold changes. This suggests that: (i) MinLibCas9 preserves enough density of sgRNAs for CRISPRcleanR to correct in an unsupervised way the copy number driven deleterious effect, and (ii) alternatively supervised approaches could also be used in case the density becomes too low. CERES is also a supervised approach, although in this instance it will be difficult to use it as it requires large sample panels since the effects are fitted across samples. In contrast, both Crispy and CRISPRcleanR work on a single sample basis. Overall, MinLibCas9 screens can be readily analysed using current tools for copy number correction.

“““

Minor points:

Figure 2: typo in Fig2b y-axis label ('precision').

”””

Thank you for reporting, this has been corrected.

“““

Supp Fig 2: are these correct? It is difficult to believe that RuleSet2 vs Median is basically uncorrelated ($R=-0.03$), while FORECasT vs Median is near-perfect (anti)correlation ($R=-0.97$).

””””

Apologies there was indeed a mistake on the assignment of the correlation coefficient labels. We have now fixed this.

““““

Supp Fig 5: what are the A/B annotations in the X axis labels?

””””

We apologise for forgetting to add this into the legend, these correspond to two different experiments,

““““

Supp Fig 7: excellent use of RMSE in scatter plots designed to show correlation. This is how everyone should do it.

””””

We thank the reviewer for the positive comment. Indeed we think this is a complementary metric to the correlation coefficient.

“““

Reviewer #3: Goncalves et al. present a minimal genome-scale CRISPR-Cas9 library with 2 sgRNAs targeting each gene. This approach will decrease costs and enable many applications. This represents a valuable resource for the functional genomics community.

”””

We thank the reviewer for believing this could be a useful resource for the scientific community.

“““

Major concerns:

The key question that the paper must address in as many ways as possible is: how can the authors ensure that by selecting for non-essential guides with the highest DNA cutting toxicity that they are not somehow selecting for unpredicted off-target activity? Is all of this analysis DNA copy number corrected? DNA copy number amplification increases the Cas9 DSB cutting effect at both on and off target sites. The authors try to examine this by looking at the deviation between guide phenotypes for guides that target the same gene but given that the original Cas9 sgRNA libraries that they are using to analyze the data only have 4-10 sgRNAs they are working with limited information on the distribution of phenotypes expected for guides with high on or off target activity which limits how robust this outlier analysis can be.

”””

We completely agree with the reviewer, unappreciated off-target activity of some sgRNAs can contribute to their selection. Nonetheless we believe our selection procedure will penalise these by:

(i) KS-score is calculated by comparing the sgRNA fold-change distribution to non-targeting sgRNAs across a large number and etiologically distinct set of cancer cell lines, with many different profiles of copy number and ploidy alterations. Hence, copy number will drive the dependency of sgRNAs in only a small set of cell lines and therefore will not contribute strongly to the KS-score;

(ii) We agree, higher number of sgRNAs would provide better estimations of JACKS scores, although large libraries are harder to screen and previous analysis supported the utility of this approach to identify potential sgRNAs that are outliers [3]. Moreover, combining

JACKS and KS-scores improves the the identification of essential genes and precision/recall of dependencies identified with the larger library (Supplementary Figure 3);

(iii) It is also important to note that the reference library, from which we perform sgRNA selection, is assembled utilising sgRNA libraries that have been already independently optimised by off-target and on-target activity;

Taken together, considering that the *in silico* downsampling and additional experimental validations using MinLibCas9 generates very consistent results compared to original and independent library screens (please also see point Reviewer #1 Major Point 4) we believe our reference library and selection procedure is enriched for sgRNAs with higher on-target activity.

““““

In this paper the authors have expected genes that should be identified but in general this is not the case in functional genomics screens that are not based on identification of essential genes (e.g. genes that modulate response to a new drug that hasn't previously been tested with a larger library). With just two sgRNAs per gene the authors should provide guidance on how they expect to call hit genes in functional genomics assays. Are they expecting people will only use the log2 fold change or Z scores relative to the negative control distribution? With only two guides most statistical tests cant be used. Ideally the authors can use existing or new data to provide an example of how to do this and how robust this can be.

””””

Reviewer point is very well taken. Reducing the number of sgRNAs per gene to obtain smaller libraries comes at the expense of having higher uncertainty on gene-level estimates.

For positive selection experiments, such as CRISPR-Cas9 screens with drug treatment, we showed on Supplementary Figure 6 that a subsampled MinLibCas9 can still robustly recapitulate the top hits found using the larger library, i.e. Project Score. Additionally, it is important to note that these experiments were not taken in consideration for sgRNA selection for MinLibCas9 and therefore is an independent assessment.

Regarding the computational approaches, similar point was raised by Reviewer #2, major point #1, and we would like to indicate the reviewer to please see our response. We added a new Supplementary Figure 9 that contains a benchmark of current tools to correct

copy number biases and call significant dependencies from CRISPR-Cas9 screens and show that these tools are readily applicable to MinLibCas9 screens, we show that:

(i) Both unsupervised and supervised approaches, CRISPRcleanR and Crispy respectively, can be used to correct copy number bias on MinLibCas9 screens. We have shown this with with MinLibCas9 on HT-29 colorectal cancer cells, which harbour several copy number alterations, including strong MYC amplifications (≥ 8 absolute copies);

(ii) Using a subsample analysis of MinLibCas9 from Project Score library, we confirmed that BAGEL can be immediately used to estimate significant gene dependencies with MinLibCas9 and these are largely concordant with Project Score across hundreds of cancer cell lines;

Overall, we believe this confirms that current CRISPR-Cas9 pipelines are readily applicable to MinLibCas9. Nonetheless, we expect that analytical adaptations of these methods to the decreased number of reagents per gene and therefore increased noise will likely make these tools more suitable for minimal libraries.

“““

Please show replicate scatter plots rather than just pearson summary statistics for Figure S5B as in Figure S8C. It is quite surprising that the authors claim coverage as low as 25x is sufficient for a genome scale screen. It had previously been shown/claimed that higher coverage in such assays is correlated with increased reproducibility ([https://urldefense.proofpoint.com/v2/url?u=https-3A_www.nature.com_articles_nrg.2017.97&d=DwIGaQ&c=D7ByGjS34AIIFgEcYw0iC6Zq7qIm8uclZFI0SqQnqBo&r=LPqRwSV8ZB--GiAMnQsNtLUNc8ghb4uIWQubaLKbxIE&m=JW_qERNjT_yhxEyHWkRV_e4dMM8tQIGS8TA2CAf6sr4&s=18yAiAmg9eRY6s_t1Og162ws1CteJGj97U_31rF3slc&e](https://urldefense.proofpoint.com/v2/url?u=https-3A_www.nature.com_articles_nrg.2017.97&d=DwIGaQ&c=D7ByGjS34AIIFgEcYw0iC6Zq7qIm8uclZFI0SqQnqBo&r=LPqRwSV8ZB--GiAMnQsNtLUNc8ghb4uIWQubaLKbxIE&m=JW_qERNjT_yhxEyHWkRV_e4dMM8tQIGS8TA2CAf6sr4&s=18yAiAmg9eRY6s_t1Og162ws1CteJGj97U_31rF3slc&e=)). Why is this data only shown for KM-12 cells? Why did the authors switch to using a pearson correlation in Figure S5B but everywhere else use a spearman? What coverage was Project Score run at? Are the authors really saying they think functional genomics screens should be run at say 50x coverage?

”””

We apologise for the confusion we did not meant to be inconsistent on the correlation metric used and changed Supplementary Figure 5b to report Spearman's Rho instead, and also as suggested we added the scatter plots to Supplementary Figure 5.

To test the impact of different levels of sgRNA coverage required separate experimental work since Project Score screens were performed using a 100x sgRNA representation. Thus, we screened at different coverage levels, lower and higher, KM-12 cancer cells. In this analysis we wanted to compare how MinLibCas9 (subsampling) performs compared to Project Score library, rather than advising on the best guide coverage to perform functional genetic screens. Additionally, for the later we believe that other metrics should be taken in consideration, e.g. replicate correlation, since the classification of essential genes from non-essential might not provide enough sensitivity to fully assess this.

“““

This manuscript is only useful/impactful to other researchers in the field if the sgRNA library is made public through addgene or some other open access source.

”””

We completely agree, we are now in the process of making the library available on Addgene.

1. Shi J, Wang E, Milazzo JP, Wang Z, Kinney JB, Vakoc CR. Discovery of cancer drug targets by CRISPR-Cas9 screening of protein domains. *Nat Biotechnol.* 2015;33:661–7.
2. He W, Zhang L, Villarreal OD, Fu R, Bedford E, Dou J, et al. De novo identification of essential protein domains from CRISPR-Cas9 tiling-sgRNA knockout screens. *Nat Commun.* 2019;10:4541.
3. Allen F, Behan F, Khodak A, Iorio F, Yusa K, Garnett M, et al. JACKS: joint analysis of CRISPR/Cas9 knockout screens. *Genome Res.* 2019;29:464–71.
4. Sanson KR, Hanna RE, Hegde M, Donovan KF, Strand C, Sullender ME, et al. Optimized libraries for CRISPR-Cas9 genetic screens with multiple modalities. *Nat Commun.* 2018;9:5416.
5. Tzelepis K, Koike-Yusa H, De Braekeleer E, Li Y, Metzakopian E, Dovey OM, et al. A CRISPR Dropout Screen Identifies Genetic Vulnerabilities and Therapeutic Targets in Acute Myeloid Leukemia. *Cell Rep.* 2016;17:1193–205.
6. Dempster JM, Pacini C, Pantel S, Behan FM, Green T, Krill-Burger J, et al. Agreement between two large pan-cancer CRISPR-Cas9 gene dependency data sets. *Nat Commun.* 2019;10:5817.
7. Behan FM, Iorio F, Picco G, Gonçalves E, Beaver CM, Migliardi G, et al. Prioritization of cancer therapeutic targets using CRISPR-Cas9 screens. *Nature.* 2019;568:511–6.
8. Pacini C, Dempster JM, Gonçalves E, Najgebauer H, Karakoc E, van der Meer D, et al. Integrated cross-study datasets of genetic dependencies in cancer [Internet]. *bioRxiv.* 2020 [cited 2020 Jul 16]. p. 2020.05.22.110247. Available from: <https://www.biorxiv.org/content/10.1101/2020.05.22.110247v2>
9. Rauscher B, Heigwer F, Breinig M, Winter J, Boutros M. GenomeCRISPR - a database for high-throughput CRISPR/Cas9 screens. *Nucleic Acids Res.* 2017;45:D679–86.
10. Lenoir WF, Lim TL, Hart T. PICKLES: the database of pooled in-vitro CRISPR knockout library essentiality screens. *Nucleic Acids Res.* 2018;46:D776–80.
11. Liu J, Srinivasan S, Li C-Y, Ho I-L, Rose J, Shaheen M, et al. Pooled library screening with multiplexed Cpf1 library. *Nat Commun.* Nature Publishing Group; 2019;10:3144.
12. Hart T, Moffat J. BAGEL: a computational framework for identifying essential genes from pooled library screens. *BMC Bioinformatics.* 2016;17:164.
13. Gonçalves E, Behan FM, Louzada S, Arnol D, Stronach EA, Yang F, et al. Structural rearrangements generate cell-specific, gene-independent CRISPR-Cas9 loss of fitness effects. *Genome Biol.* 2019;20:27.
14. Iorio F, Behan FM, Gonçalves E, Bhosle SG, Chen E, Shepherd R, et al. Unsupervised correction of gene-independent cell responses to CRISPR-Cas9 targeting. *BMC Genomics.* 2018;19:604.

Second round of review

Reviewer 1

The authors have answered most of the questions and clarified confusions. Major concerns have been addressed properly.

Reviewer 3

The authors have addressed all of my concerns.